# Novel Mixed Cancer-Cell Models Designed to Capture Inter-Patient Tumor Heterogeneity for Accurate Evaluation of Drug Combinations

**DOI:** 10.3390/ijms27010413

**Published:** 2025-12-30

**Authors:** Sampreeti Jena, Daniel C. Kim, Adam M. Lee, Weijie Zhang, Kevin Zhan, Radwa M. Elmorsi, Yingming Li, Scott M. Dehm, R. Stephanie Huang

**Affiliations:** 1Department of Experimental and Clinical Pharmacology, University of Minnesota, Minneapolis, MN 55455, USA; jenax004@umn.edu (S.J.); leeam@umn.edu (A.M.L.); zhan8663@umn.edu (K.Z.); radwa.elmorsy@med.tanta.edu.eg (R.M.E.); 2Department of Bioinformatics and Computational Biology, University of Minnesota, Minneapolis, MN 55455, USA; zhan6385@umn.edu; 3Department of Laboratory Medicine and Pathology, University of Minnesota, Minneapolis, MN 55455, USA; lixxx354@umn.edu (Y.L.); dehm@umn.edu (S.M.D.)

**Keywords:** tumor heterogeneity, mixed-cell model, preclinical screening, drug combinations

## Abstract

Disease heterogeneity across a diverse patient cohort poses challenges to cancer drug development due to inter-patient variability in treatment responses. However, current preclinical models fail to depict inter-patient tumor heterogeneity, leading to a high failure rate when translating preclinical leads into clinical successes. We integrated the expression profiles of prostate cancer (PC) lines and castration-resistant PC (CRPC) patient tumors to identify cell-lines that transcriptomically match distinct tumor subtypes in a clinical cohort. Representative cell-lines were co-cultured to create “mixed-cell” models depicting inter-patient heterogeneity in CRPC, which were employed to assess drug combinations. When drug combinations previously tested in CRPC clinical cohorts were assessed to establish proof of concept, in vitro responses measured in our models concurred with their known clinical efficacy. Additionally, novel drug combinations computationally predicted to be efficacious in heterogeneous tumors were evaluated. They demonstrated preclinical efficacy in the mixed-cell models, suggesting they will likely benefit heterogeneous patient cohorts. Furthermore, we showed that the current practice of screening cell-lines/xenografts separately and aggregating their responses, failed to detect their efficacy. We believe that the application of our models will enhance the accuracy of preclinical drug assessment, thereby improving the success rate of subsequent clinical trials.

## 1. Introduction

Cancer is a heterogeneous disease at the genetic, epigenetic, and phenotypic levels due to the stochastic and dynamic nature of its origin and evolution. Tumor heterogeneity is manifested in several forms. Inter-patient heterogeneity occurs on the population level and refers to the molecular and phenotypic diversity across distinct patient tumors. On the other hand, intra-tumoral heterogeneity refers to the cellular subpopulations harboring distinct molecular and phenotypic signatures in a single patient tumor [1]. Of these, inter-patient heterogeneity is typically more pronounced and clinically prevalent and is therefore the focus of the current investigation. Inter-patient heterogeneity gives rise to variability in therapeutic responses among patients. Patients that are intrinsically resistant to anticancer therapies exhibit inferior clinical outcomes, eventually leading to mortality [2,3]. Varying treatment sensitivities among individuals also pose challenges to the development of novel therapies [4].

Currently, more than 80% of novel anti-cancer compounds which show promising preclinical results do not clear Phase II clinical trials [5,6]. Typically, preclinical testing has been carried out in models such as individual cancer cell-lines or patient-derived cells, in vitro and in vivo. The selection of these models is primarily empirical and based on availability. This practice fails to reflect the variability in drug responses across diverse patient cohorts and therefore produces inaccurate results. Furthermore, the systematic evaluation of FDA-approved combinatorial therapies that have been tremendously beneficial in improving treatment outcomes and mitigating therapy resistance, has revealed that their clinical efficacy is linked to heterogeneity [7,8]. By the principle of independent drug action (IDA), each component drug in the combination (combo) independently targets a distinct variant/subtype of the heterogeneous disease. The current method for preclinical drug testing interrogates one homogeneous model at a time and then aggregates the results from these models to estimate overall combo efficacy. This practice cannot accurately assess drug combos whose efficacy is attributable to IDA. To overcome these issues, we were motivated to develop new preclinical models that capture inter-individual tumor heterogeneity within diverse patient cohorts. Our rationale is that by advancing only drug/drug combos that are preclinically validated in these models, we will increase the efficiency and success rate of their subsequent clinical validation. Improving preclinical models has always been a high priority in the field of cancer biology. For example, advances in tissue engineering, such as new biomaterials and microfluidics, have facilitated improved culture quality and reproducibility in ex-vivo models [9,10]. More recently, co-culturing tumor cells with cancer-associated fibroblasts, immune and endothelial cells in organoid models, achieved the complex cellular architecture of patient tumors [11]. Advances have also been made in improving PDX models which allow the interrogation of drug response in the presence of an intact tumor microenvironment (TME), under physiological conditions [12,13,14,15,16]. Yet, the majority of these models focus on recapitulating tumor cell-extrinsic features, mainly its interaction with the TME. A model that replicates the intrinsic genetic and phenotypic heterogeneity in cancer cells derived from distinct patients is *absent*. To bridge this gap, the work presented here focuses on creating novel in vitro models that capture the complex and diverse transcriptomic landscape of cancer cells in a large clinical cohort, for utility in preclinical drug combo testing. We hypothesized that a mixture (co-culture) of cancer cell-lines representing genetically distinct tumor variants harbored in a diverse patient cohort can broadly encompass the scope of inter-patient heterogeneity in a specific cancer type. It is worth noting that pooling cancer cell-lines and using them for drug screens has been carried out by the Broad Institute PRISM platform, where up to 578 cancer cell-lines were DNA-barcoded and combined for collective screening across a panel of 4518 drugs [17,18]. However, since the end goal of the PRISM project was to enable high throughput screening of large drug libraries, their rationale for cell-line pooling was based on the similarity in cell growth rates and/or tissue origin. Conversely, in this study, rationally selected cancer cell-lines were combined with the goal of replicating disease-specific genetic heterogeneity in patient tumor cells from a diverse cohort.

In this study, we chose to focus on castration resistant prostate cancer (CRPC), a highly lethal stage of prostate cancer that accounts for nearly all PC mortality. CRPC is a classic example of a heterogeneous disease, with subtypes ranging from an adenocarcinoma phenotype to the highly refractory neuroendocrine differentiated PC (NEPC) [19]. In CRPC, intrinsic and acquired therapy resistance is common [20], and very limited treatment options are available [21]. Using a data-guided approach, we integrated transcriptomic datasets from large CRPC patient cohorts with PC cell-lines to identify representative cell-lines that genetically resemble distinct patient tumor subtypes (Figure 1). These rationally selected cell-lines were pooled to create mixed-cell models reflecting heterogeneity in the following two distinct settings in CRPC: (1) cohorts of unstratified CRPC patients and (2) cohorts of CRPC patients with a prior history of taxane exposure who progressed to a taxane-resistant phenotype, a common occurrence in the clinical management of CRPC. To demonstrate proof of concept, drug combos which either demonstrated efficacy or failed in CRPC clinical trials, were tested in the newly developed mixed-cell models. Additionally, they were employed to preclinically assess novel drug combos likely to confer efficacy in heterogeneous CRPC cohorts, on the basis of IDA.

## 2. Results

### 2.1. Representing Inter-Patient Heterogeneity in CRPC Tumors

#### 2.1.1. Creation of a Mixed-Cell Model to Represent Inter-Patient Heterogeneity in CRPC Patients

We integrated “tumor-only” expression profiles imputed from the bulk transcriptomes of 208 CRPC patients from the Stand Up to Cancer East Coast Prostate Cancer study (SU2C/PC-EC) [22] and PC cell-lines collected from the Cancer Cell-line Encyclopedia (CCLE) [23] and the Genomics of Drug Sensitivity in Cancer (GDSC) [24]. An unsupervised hierarchical clustering was conducted to segregate samples based on their transcriptomic similarity. The dendrogram from the analysis (Figure 2A) revealed four primary clusters, each containing cell-line and patient tumor samples. Cell-lines with similar molecular and clinical backgrounds (e.g., androgen dependency, stage of progression, and clinical aggressiveness) clustered together, lending credibility to our pipeline. More importantly, co-clustered patient specimens also shared the genetic and phenotypic characteristics of the representative cell-lines, such as AR and NEPC signature scores calculated from the expression levels of known genetic markers [24] (blue and red bars in Figure 2A). A single cell-line was selected from each primary cluster (barring cluster 1) as its transcriptomic surrogate to be incorporated in the model as follows: DU145, an androgen receptor (AR)-negative cell-line; 22RV1, a cell-line that co-expresses AR and constitutively-active AR variants; and R1D567, a cell-line model that lacks AR but instead expresses constitutively active AR variant (ARv567es) [25]. Collectively, these cell-lines encompass a broad spectrum of the pathology and molecular biology exhibited by CRPC patient tumors. Notably, cluster 1, with the highest NEPC signature score among all patient specimens, contained a single cell-line, NCI-H660. H660, a known model of NEPC [26], grows in very different culturing conditions than the other three lines and therefore was not included in the model. To ensure robust selection of cell-lines and consistency across different CRPC cohorts, we repeated the clustering analysis with an independent CRPC patient dataset from the Stand Up to Cancer West Coast Prostate Cancer (SU2C/PC-WC) study (Appendix A). Similar results were obtained with 22RV1, R1D567, DU145, and H660 being assigned into separate patient clusters. Selected cell-lines were tagged with unique fluorescent labels (DU145:RFP, 22RV1:BFP, and R1D567:GFP) and mixed at a ratio (1:2:2) proportional to their relative doubling times, to account for the differences in unperturbed growth rates. As shown in Figure 2B, during 96 h of growth monitoring in the pooled co-cultures, each PC line proliferated as expected in the presence of the other lines, following the sigmoidal growth pattern.

#### 2.1.2. Evaluation of Clinically Efficacious and Inefficacious Drug Combos to Establish Proof of Function

To demonstrate the functional utility and accuracy of this model, we tested two clinically validated drug combos, one of which achieved superior clinical efficacy over either monotherapy (positive proof of concept) and another which failed to do so in PC patient cohorts (negative proof of concept). Currently, docetaxel is a standard of care (SoC) therapy in the treatment of CRPC. Previous preclinical studies have shown that the addition of a tyrosine kinase inhibitor, dasatinib, to docetaxel further reduced tumor growth, migration, and bone metastasis in mouse models compared to docetaxel alone [27,28]. Note that the mouse xenograft models employed in these studies incorporated a single PC line, C4-2B. Moreover, dasatinib and docetaxel combo therapy was found to be well-tolerated in a phase I trial [29]. However, in a phase III, double-blind, randomized controlled trial of 1522 men who received docetaxel plus prednisone along with either dasatinib or placebo, no difference was observed between the median overall survival times of the two arms (HR 0.99, 95.5% CI 0.87–1.13) [30]. Therefore, docetaxel + dasatinib was selected to be tested in our new mixed-cell model to establish negative proof of concept. Growth analysis of the constituent PC lines in the co-cultures (Figure 3A–C) revealed that inhibition of DU145 (Figure 3A) and 22RV1 (Figure 3B) CRPC subtypes was driven by docetaxel, with no additional benefit being conferred by dasatinib. In R1D567 (Figure 3C), a slight yet significant improvement in efficacy was achieved by the combo (*p* = 0.027), as indicated by the difference between the measured combo efficacy and the dashed lines denoting the effect of the most efficacious monotherapy. However, this marginal improvement was imperceptible in the composite tumor mixture where the combo failed to enhance anti-tumor efficacy over the most potent monotherapy, docetaxel (Figure 3D). Therefore, the in vitro efficacy of the drug combo measured in the mixed-cell model was consistent with the Phase III clinical trial results. An identical experimental approach and procedure were adopted to evaluate the drug combo of docetaxel and vinorelbine as positive proof of concept (Figure 4). This combo has demonstrated preclinical potency in both androgen-dependent and androgen-independent PC cell-lines. Clinically, the combo improved the objective response rate relative to current CRPC therapy regimens [31,32]. When assessed in our model, the combo achieved greater inhibition of the component cell-lines (Figure 4A–C) as well as of the tumor mixture (Figure 4D) compared to either single-agent drug. Once again, the mixed-cell model could successfully reproduce the therapeutic outcomes observed clinically in patient studies.

#### 2.1.3. Nomination and Preclinical Validation of Novel Drug Combos for CRPC

Next, we employed the mixed-cell model to experimentally evaluate a number of novel drug combos that were computationally nominated based on the principle of independent drug action (IDA). Using IDACombo [33], a computational method that leverages measured monotherapy responses in cancer cell-lines to predict drug combo efficacy in heterogeneous tumors, we imputed the IDACombo scores between docetaxel and 510 other drugs. Note that higher IDACombo scores imply greater expected efficacy of the combo in heterogeneous tumors. Combos were ranked in decreasing order of their predicted efficacies. The top hit with the highest IDACombo score that emerged from this analysis was the combo of daporinad, a Nicotinamide Phosphoribosyltransferase (NAMPT) inhibitor, and docetaxel. Figure 5A illustrates the imputed combo efficacies generated by IDACombo for docetaxel and daporinad combined (pink dots) at various doses. As indicated by the separation between the pink dots and the grey plane (denoting the effects of the most efficacious monotherapy), the majority of all possible combos between the two drugs are predicted to produce a higher potency than the best monotherapy. To assess the therapeutic efficacy of this drug combo, we tested it in the three-cell-line mixed-cell model (DU145:RFP + R1D567:GFP + 22RV1:BFP). While R1D567 was comparably sensitive to both drugs (Figure 5D), DU145 (Figure 5B) and 22RV1 (Figure 5C) were primarily inhibited by daporinad and docetaxel, respectively, as indicated by the overlap between the measured combo efficacy and the dashed lines denoting the effect of the most efficacious monotherapy. No statistical difference was observed between the effects of the combo and the most effective single-agent therapy (black dashed lines) within each component cell-line, suggesting the absence of synergistic/antagonistic interactions between the drugs. Analysis of the total tumor mixture (Figure 5E) revealed that the overall efficacy of the combo was superior to that of the monotherapies (2-way ANOVA *p* < 0.05) and resulted in greater inhibition of the tumor mixture. However, if the drug combo was evaluated in each cell-line individually and their measured responses aggregated for each treatment condition (in accordance with common pharmacological practice), no statistically significant difference was observed between the efficacies of docetaxel monotherapy and the combo (Figure 5F).

### 2.2. Development of Mixed-Cell Model to Capture Inter-Patient Heterogeneity in Taxane-Resistant CRPC Tumors

Aside from intrinsic therapeutic resistance, acquired drug resistance also poses significant challenges in the management of CRPC. Underlying molecular mechanisms driving acquired drug resistance also vary among distinct tumor subtypes. Therefore, there is a need to create models that represent the heterogeneity in SoC-resistant tumors to identify efficient drug(s) for their treatment. We first sought to evaluate the molecular nature and degree of heterogeneity in SoC-resistant CRPC tumors. Subsequently, we rationally selected and pooled representative cell-lines to create a new mixed-cell model that captures inter-patient heterogeneity in taxane-resistant CRPC patients and utilized them for testing novel drug combos.

#### 2.2.1. Understanding Heterogeneity in Docetaxel-Resistant CRPC

The molecular mechanisms driving acquired docetaxel resistance are diverse. We first developed a collection of docetaxel-resistant PC cell-lines by chronically exposing the sensitive parent lines to sub-lethal dosages of docetaxel. The resistant PC cell-lines thus established exhibited on average 10–100 times greater IC50s for docetaxel when compared to the parent lines [34,35]. Subsequently, RNA sequencing was performed to obtain the transcriptomic profiles of docetaxel-sensitive and docetaxel-resistant PC cell-lines. Our findings suggest that molecular perturbations induced by acquired docetaxel resistance are indeed specific to the tumor subtype. As shown in Figure 6A, by performing GSEA pathway enrichment analysis on differentially expressed genes between docetaxel-sensitive (doceS) and docetaxel-resistant (doceR) clones, separately for 22RV1 and DU145 (representing distinct CRPC subtypes (Figure 2A)), differential patterns of molecular pathway enrichment were revealed. Specifically, upregulation of the GF/RTK/Ras/ERK signaling pathway and ABC transporters was observed in the doceR 22RV1. However, the same GF/RTK/Ras/ERK pathway was significantly downregulated in doceR DU145. Conversely, the doceR DU145 exhibited enrichment of the PI3K/Akt/mTOR pathway, which was absent in 22RV1. Indeed, previous publications have associated these mechanisms with the development of docetaxel resistance in PC [36,37,38,39,40,41]. In light of these findings, we reasoned that combinatorial therapies targeting these distinct molecular vulnerabilities are more likely to be effective in heterogeneous docetaxel-resistant CRPC tumors.

#### 2.2.2. Creation of a Mixed-Cell Model to Represent Heterogeneous Docetaxel-Resistant CRPC

We integrated transcriptome data from 79 CRPC tumor specimens derived from patients who had previously received taxane therapy in the SU2C/PC-EC study and a collection of PC cell-lines including both parent (doceS) and docetaxel-resistant (doceR) lines for clustering analysis. We observed that the cell-lines and patient samples segregated into four primary clusters (Figure 6B), with the DU145, R1D567, and 22RV1 cell-lines being assigned to separate clusters. The doceS and doceR clones derived from the same cell-lines were clustered together, indicating that transcriptomic variability among different patient tumors is larger than changes induced by docetaxel exposure. It may be further inferred that inter-patient genetic variability in taxane-exposed patients can be largely attributed to differences in the intrinsic molecular composition (tumor molecular background) instead of variations induced by taxane therapy. Consequently, to best represent inter-patient heterogeneity in docetaxel-resistant CRPC, we selected three doceR PC lines, one from each patient cluster in Figure 6B. These lines were labeled (as doceR 22RV1-GFP, doceR DU145-BFP, and doceR R1D567-RFP) and pooled to create the new mixed-cell model. This model was then employed in evaluating novel drug combos that may help overcome docetaxel resistance in a heterogeneous patient cohort.

#### 2.2.3. Nomination and Preclinical Validation of Novel Drug Combos for Docetaxel-Resistant CRPC

To discover candidate drug combo(s) that are effective in treating doceR CRPC tumors, we employed oncoPredict, a computational tool trained on the expression data of cancer cell-lines and proven to accurately project drug responses in patient tumor specimens [42]. We hypothesized that a combo of drugs that are separately efficacious in different component cell-lines of the doceR mixed-cell model will collectively enhance therapeutic efficacy in the tumor mixture. MEK inhibitors and NAMPT inhibitors were predicted to be the most effective therapies against doceR 22RV1 and DU145, respectively. These drug nominations were concordant with results from the pathway enrichment analysis where the Ras/Raf/MEK/ERK and NAMPT pathway signatures were found to be significantly upregulated in doceR 22RV1 and DU145, respectively (Figure 6A). In vitro drug testing performed individually in these cell-lines corroborated the computational predictions. While daporinad, an NAMPT inhibitor, exhibited a significantly higher effect in doceR (compared to DoceS) DU145 cells, selumetinib, a MEK inhibitor, selectively inhibited doceR 22RV1 (Appendix A).

The doceR mixed-cell models (doceR 22RV1-GFP, doceR DU145-BFP, and doceR R1D567-RFP) were treated with the combo of selumetinib and daporinad, single agent drugs or the vehicle control (DMSO-containing media). Normalized growth areas of the constituent doceR lines and the cellular mixture as well as representative 3-channel fluorescence images after 5 days of treatment are shown in Figure 7. It is evident that growth inhibition in doceR DU145 and R1D567 was conferred by daporinad (Figure 7A,C). This was further confirmed in the microscopy images from the depletion of both subpopulations and the preferential enrichment of doceR 22RV1 (green) when treated with daporinad only (Figure 7F). 22RV1, on the other hand, was selectively inhibited by selumetinib, irrespective of the presence of daporinad, leading to its preferential depletion when treated with selumetinib alone (Figure 7B,G). The combo, however, could concurrently inhibit all three subpopulations, and therefore enhanced overall therapeutic efficacy compared to either monotherapy in the tumor mixture (Figure 7D,H).

## 3. Discussion

In this study, we aimed to develop preclinical models that capture inter-patient tumor heterogeneity, for which the transcriptomic similarity between CRPC patient tumors and PC cell-lines was comprehensively assessed. This step allows us to rationally select PC cell-lines that resemble genetically distinct patient tumor subtypes. By evaluating drug combos of known clinical efficacy, we demonstrated that our models can accurately replicate their observed clinical efficacy. Moreover, we applied these models to evaluate novel drug combos computationally prioritized on the basis of IDA. To our knowledge, this is the only available preclinical model to date that enables testing of IDA-based drug combos. For drug testing, the pooled co-cultures were treated as a composite heterogeneous tumor where therapeutic efficacy was commensurate with overall inhibition of the tumor mixture. This strategy represents a departure from the more traditional approach of testing drugs on individual cancer cell-lines or PDXs, one model at a time, and aggregating their individual responses, a practice that was shown to be ineffective in accurately assessing these drug combos.

Several novel combinatorial therapies that were computationally predicted to be effective in heterogeneous CRPC cohorts were prioritized and preclinically validated in our new mixed-cell models. The combo of docetaxel and the NAMPT inhibitor daporinad was found to achieve greater anti-tumor efficacy than either monotherapy treatment in the heterogeneous cellular mixtures. Recent studies have shown that NAMPT is overexpressed in PC cell-lines and patient tumor samples compared to healthy prostate tissue [43]. NAMPT promotes prostate tumorigenesis through metabolic and transcriptional reprogramming and enables survival under oxidative and chemotherapeutic stress (resistance) [44]. The efficacy of NAMPT inhibitors in neuroendocrine PC (NEPC), one of the most advanced and lethal forms of PC, has been previously reported in the literature [45,46]. In our study, we showed that at doses far below the IC50s for either docetaxel or daporinad (Appendix A), the predicted combo effects were already greater than the most effective monotherapy, suggesting that combining low doses of these drugs can achieve superior tumor inhibition compared to regular doses of either single agent. This is a major advantage since past clinical trials with NAMPT inhibitors were halted due to dose-limiting toxicity [47,48]. Once validated in vivo, this new combo therapy could allow clinicians to administer NAMPT inhibitors at very low dosages, minimizing toxicity issues. Importantly, if the drug treatments were performed in any single-component cell-line or individually in all three of them and their responses aggregated (as per the norm in pharmacology), the combo would be deemed ineffective due to the apparent absence of drug synergy or additivity within each cell-line and the large variation in the aggregated responses. However, when the pooled cellular mixture was analyzed as a composite entity analogous to a cohort of heterogeneous CRPC tumors, the combo is predicted to confer clinical efficacy by benefiting distinct patient groups. Indeed, when the predicted sensitivities of patients from the SU2C/PC-EC cohort to docetaxel and daporinad monotherapies were stratified by their cluster assignment (obtained from the clustering analysis), a pattern of intrinsic collateral sensitivity between the two drugs emerged (Appendix A). In other words, patient groups that responded the least to docetaxel were predicted to be most sensitive to daporinad and vice versa. These observations suggest that the drug combo likely confers therapeutic efficacy through IDA, which according to the recent literature could account for the performance of most clinically successful drug combos [7].

It is well established that clonal selection or genetic/epigenetic reprogramming induced by chronic drug exposure alters drug sensitivities in tumors [49]. Earlier studies have shown that therapy resistance can be acquired through the dysregulation of various molecular pathways [50]. This was also supported by our data. We nominated a two-drug combo, selumetinib + daporinad, with the goal of overcoming docetaxel resistance in a heterogeneous patient cohort. When tested in a mixed-cell model composed of three doceR PC lines, we observed that the constituent drugs in the combo targeted distinct cellular subpopulations in the mixture, independently. Collectively, the combo concurrently inhibited all three subpopulations and therefore enhanced overall therapeutic efficacy compared to the individual monotherapies. Previously, it has been reported that daporinad exhibited preclinical efficacy in doceR DU145 and PC3 lines [40], while selumetinib was shown to inhibit DU145 and PC3 growth, synergistically, in combo with Akt and PI3Kβ/δ inhibitors [51]. However, the combo of daporinad and selumetinib has not been investigated before in doceR CRPC models. Similar to the previous example (docetaxel + daporinad), this combo conferred therapeutic benefit through IDA by targeting distinct doceR CRPC subtypes in the model, which would not be apparent if the constituent cell-lines were interrogated one at a time. Thus, we demonstrated the utility of these novel mixed-cell models in validating IDA-based drug combos that would be rejected if traditional preclinical models or drug screening practices were employed. However, as exemplified by the docetaxel + vinorelbine combo evaluation, our models can also identify drug combos that produce efficacy through synergy, wherein the combo achieved greater inhibition than either single-agent therapy in each constituent cell-line. The presence of synergy was confirmed by the synergy scores (Appendix A) calculated using the Bliss independence model [52]. 

The goal and the novelty of this study is the representation of tumor-intrinsic inter-patient diversity in diverse clinical cohorts for preclinical drug evaluation, a topic that has been frequently overlooked in the field. However, it is important to acknowledge the limitations associated with our models. Firstly, our 2D cultures do not recapitulate tumor 3D morphology and the TME. Our rationale is that, despite the new findings of TME contributions to drug sensitivity, our results continue to support inter-patient differences in tumor molecular composition as the main driver of variability in drug response. Moreover, while there are substantial efforts in the field as of today to explore the role of TME, the impact of intrinsic inter-patient heterogeneity (genetic and phenotypic variability in tumor cells) has been largely overlooked, which remains the main focus of our work. In the future, the integration of our models representing tumor-intrinsic heterogeneity with these state-of-the-art 3D models would enable a more wholistic depiction of the human disease. We selected and combined three PC cell-lines to broadly depict inter-patient heterogeneity and to assess drug combination efficacy in a diverse cohort. Each cell-line was selected to represent a primary patient cluster (color highlighted in Figure 2A) in the mixed-cell model. However, we were unable to include the NCIH660 cell-line, representative of patient cluster 1, characterized by a high NEPC and low AR scores (colored black in the dendrogram in Figure 2A), due to its requirement of specialized culturing conditions. Therefore, findings generated using the mixture model with three PC cell-lines on the efficacy of various drug combinations may not be applicable to CRPC patients with the neuroendocrine phenotype. In the future, applying the same clustering principle and using additional labeling strategies that allow a higher degree of multiplexing, one can develop mixed-cell models containing higher numbers of representative cell-lines to capture greater complexity and diversity. In this study, we used transcriptomic data instead of proteomic for practical reasons, as the genomic and transcriptomic profiles of human tumors are more readily available. Down the road, when proteomic data are accessible for large patient cohorts, they can and should be considered for comparison with and selection of the representative cell-lines. Finally, the preclinical evaluation of the novel drug combinations in our models is only the first step towards their potential transition into the clinic. Additional investigation into the mechanism of action at the protein level is warranted. In our case, by using cell-lines with publicly available molecular information and known in vitro drug sensitivities, we can generate hypotheses more easily towards the underlying mechanisms.

## 4. Materials and Methods

### 4.1. Cell Culture and Reagents

CWR-R1-D567 (Cat. # EMN028-FP, RRID: CVCL_ZC61) cells were derived by deleting AR exons 5–7 through transcription activator-like effector nuclease (TALEN)-mediated genome engineering in CWR-R1, a prostate carcinoma epithelial cell-line derived from the recurrent CWR22 human xenograft [25]. DU145 (Cat. # HTB-81, RRID: CVCL_0105) and 22RV1 (Cat. # CRL-2505, RRID: CVCL_1045) prostate cancer cell-lines were obtained from the American Type Culture Center (ATCC, Manassas, VA, USA). All cell-lines were grown using the RPMI 1640 medium (Thermo Fisher Scientific, Waltham, MA, USA), supplemented with 10% fetal bovine serum (FBS) (Gibco, Thermo Fisher Scientific) and maintained at 37 °C with 5% CO_2_. They were periodically monitored for mycoplasma using the Universal Mycoplasma Detection Kit following the manufacturer’s protocol (ATCC). Drugs used for in vitro testing, daporinad (FK866; CAS No. 658084-64-1), docetaxel (CAS No. 114977-28-5), dasatinib (BMS-354825; CAS No. 302962-49-8), vinorelbine ditartrate (CAS No. 125317-39-7), and selumetinib (AZD6244; CAS No. 606143-52-6), were obtained from MedChem Express (Monmouth Junction, NJ, USA). A second stock of docetaxel (CAS No. 114977-28-5) was purchased later from Selleck Chemicals (Houston, TX, USA). Drugs were dissolved in dimethylsulfoxide (DMSO) at the appropriate stock concentrations.

### 4.2. Clustering Patient and Cell-Line RNA-Seq Datasets

Transcriptomic datasets for CRPC patient biopsies from the SU2C/PCF clinical study were downloaded from the cBioportal database (http://cbioportal.org (accessed on 10 January 2023). To infer tumor-cell-specific expression data from bulk RNAseq gene expression profiles, we employed the CIBERSORTx online analysis platform (https://cibersortx.stanford.edu/) [53]. The reference signature matrix for tumor cells and non-tumor cell types was generated from single-cell RNAseq (scRNAseq) expression profiles of human CRPC biopsies [24]. This dataset uploaded by Chan et al. was obtained from the Gene Expression Omnibus (GEO) database (GSE210358). In each run, the bulk RNAseq matrices served as the “mixture” file, while the signature matrix was used as the “sigmatrix” file. Batch correction was set to “S-mode” to account for the scRNA matrix generated from the 10X platform. The “subsetgenes” parameter was configured to a file containing the intersection of gene symbols between bulk and scRNA gene expression matrices. The imputed “tumor-only” expression profiles were integrated with PC cell-line datasets and the ComBat functionality from the sva package (version 3.48.0) was implemented to remove batch effects prior to clustering. A subset of 5000 genes exhibiting the most variability among the patient samples was created from the integrated gene expression matrix. Agglomerative hierarchical clustering was applied using the Ward’s method to group samples.

### 4.3. Permanent Cellular Labeling with Lentiviral Transduction

Third generation lentiviral stocks of concentration >10^8^ TU/mL containing vectors pLV[Exp]-Puro-CMV > TagBFP (Vector ID: VB900088-2395stn), pLV[Exp]-Puro-CMV > EGFP (Vector ID:VB900088-2219pdm) and pLV[Exp]-Puro-CMV > TurboRFP (Vector ID:VB220613-1212qvs) were purchased from VectorBuilder Inc. (Chicago, IL, USA). Cells were plated at a density of approximately 400,000 cells/well in a 6-well plate in 1 mL of complete growth media containing 5 µg/mL polybrene. Viral titers ranging from 5 to 20 μL were added per well and incubated for 48 h. The viral load and incubation times were optimized to obtain the highest possible labeling efficiency without impacting cellular physiology, ascertained by comparing the growth rates of labeled and unlabeled cells. Cells were then treated with 2 µg/mL puromycin for 5–10 days to select for cells that robustly express the fluorescent proteins until stable colonies appeared. Transduced cells were expanded and monitored for up to two passages.

### 4.4. Longitudinal Measurement of Cellular Proliferation in the Mixed-Cell Model

Pooled mixtures of TagBFP-, eGFP-, and TurboRFP-labeled cells were seeded in fibronectin-coated 24-well plates (Corning^®^ BioCoat^®^, Product No. 354411) at a density of 50,000 cells per well. Wells were treated with drug combos, single-agent drugs, or the vehicle control (DMSO + media). Plates were imaged using the Cytation™ Cell Imaging Multi-Mode Reader (BioTek Instruments, Winooski, VT, USA) in DAPI, GFP, RFP, and brightfield channels. Automatic background-flattening parameters were used to remove background fluorescence. To identify tumor cells, primary masks were implemented and the cell surface area in each channel was calculated using a pixel-intensity threshold of 5000 and object size limit of 5–1000 µm. The growth area of each component cell-line was measured at the start of treatment and after 5 days of exposure. For each well, the cellular area at the treatment endpoint was divided by the area recorded at the treatment start to normalize for differences in cell seeding densities across wells. Normalized growth areas were summed to determine the growth/proliferation of the tumor mixture. As normalized proliferation values are continuous, mutually independent (both within and across groups) and can be assumed to exhibit similar distribution patterns between groups (may not be normal), the Mann–Whitney U test was employed to assess statistical differences between the treatment conditions. A total of 6 technical replicates were generated per treatment condition since the minimum sample size for the Mann–Whitney U test is generally considered to be 5.

### 4.5. Prediction of Combo Efficacy with IDACombo

Monotherapy drug responses of 887 pan-cancer cell-lines to 544 drugs were loaded from the CTRPv2 database onto the IDACombo web application (https://www.oncotherapyinformatics.org/idacombo/ (accessed on 1 December 2022)) to train the algorithm. Efficacy predictions were generated for combos of docetaxel and 510 other drugs across a range of concentrations between 0 and Csustained concentrations (maximum plasma concentration achieved at least 6 h after drug administration). The highest predicted IDAComboscore at a fixed concentration of docetaxel was recorded for each drug pair, where a higher score indicates higher combo efficacy. Drug pairs were ranked by their IDACombo scores.

### 4.6. Imputing Monotherapy Responses in Docetaxel-Sensitive and Docetaxel-Resistant Cell-Lines

Gene expression microarray data of parental doceS prostate cancer cell-lines (DU145 and 22RV1) and their docetaxel-resistant (doceR) clonal derivatives were downloaded from the Gene Expression Omnibus (GEO) with the accession number GSE36135. Cancer cell-line (CCL) drug responses were obtained from the Cancer Therapeutics Response Portal Version 2 (CTRPv2). CCL gene expression data were downloaded from the Broad Institute’s Cancer Cell-Line Encyclopedia (CCLE). For each tested drug in CTRPv2, we built a regression model using the R package oncoPredict (version 1.2). CCL sensitivities in the format of area under the dose–response curve (AUC) were used as response variables, where a lower value of AUC indicates a higher sensitivity to a drug. Transcriptomics of available CCLs were used as predictors. After model fitting, estimated coefficients of genes were applied to the gene expression profiles of doceS and doceR DU145 and 22RV1 to impute single-drug sensitivities to 334 drugs in these two clonal variants. Statistical analyses (Student’s *t* test and ANOVA) were performed to select drugs exhibiting significant differential sensitivity in the docetaxel-resistant clones.

### 4.7. Pathway Enrichment Analysis

Gene Set Enrichment Analysis comparing doceR and doceS clones was performed separately for 22RV1 and DU145 using GSEA implemented in Java GSEA application, version 2.0. A total of 50 hallmark as well as 619 KEGG_MEDICUS and 186 KEGG_LEGACY gene sets from the C2: canonical pathways collection in the Molecular Signature Database (http://www.gsea-msigdb.org/gsea/msigdb/index.jsp (accessed on 8 December 2023)) were analyzed. The detailed GSEA parameters are as follows: number of permutations: 1000; permutation type: gene set; metric for ranking genes: Diff_of_Classes; enrichment statistic: weighted; and gene set size limits: 15–500. Normalized Enrichment Scores and adjusted FDR values were obtained for each sample.

## 5. Conclusions

In summary, we developed a collection of preclinical models that reflect inter-patient heterogeneity in two distinct clinical contexts of CRPC as follows: (1) in molecularly or clinically unstratified patients and (2) in patients with prior taxane exposure. Additionally, novel drug combos with potential efficacy in diverse clinical cohorts were successfully validated in our mixed-cell models. Further in vivo evaluation of these drug combos in animal models as well as in clinical cohorts is warranted. The development of these models will be complementary to existing preclinical models that recapitulate the tumor-extrinsic microenvironment. Their application in preclinical drug screening will enable accurate and high-throughput evaluation of novel therapies and increase the success rate of advancing preclinical drug leads into the clinical stage.

## Figures and Tables

**Figure 1 ijms-27-00413-f001:**
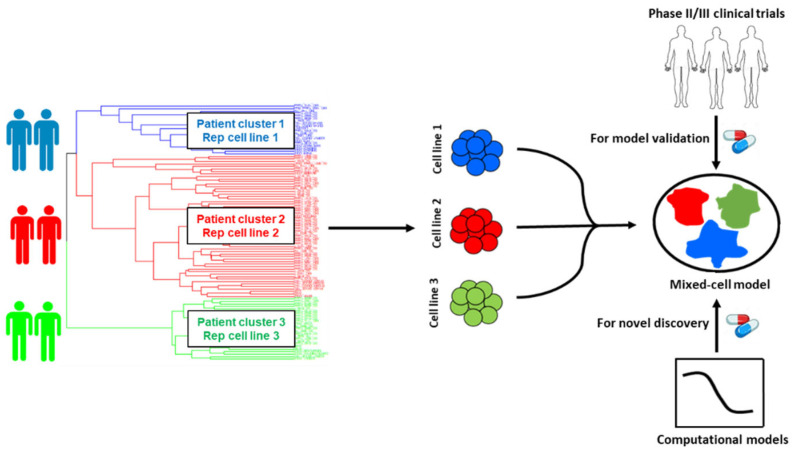
Schematic illustration of the workflow for the selection of cancer cell-lines for mixed-cell models depicting inter-patient tumor heterogeneity. Patient and cell-line RNA-seq datasets were integrated and clustered to identify cell-lines that are representative of distinct patient tumor subtypes (clusters). Selected cell-lines were fluorescently labeled and pooled to create the mixed-cell model. Combinations with known clinical efficacy in prostate cancer trials were tested to establish proof of concept and the functional accuracy of the model. Novel drug combinations predicted to be efficacious in heterogeneous patient cohorts were prioritized by implementing computational models and preclinically validated using the mixed-cell model.

**Figure 2 ijms-27-00413-f002:**
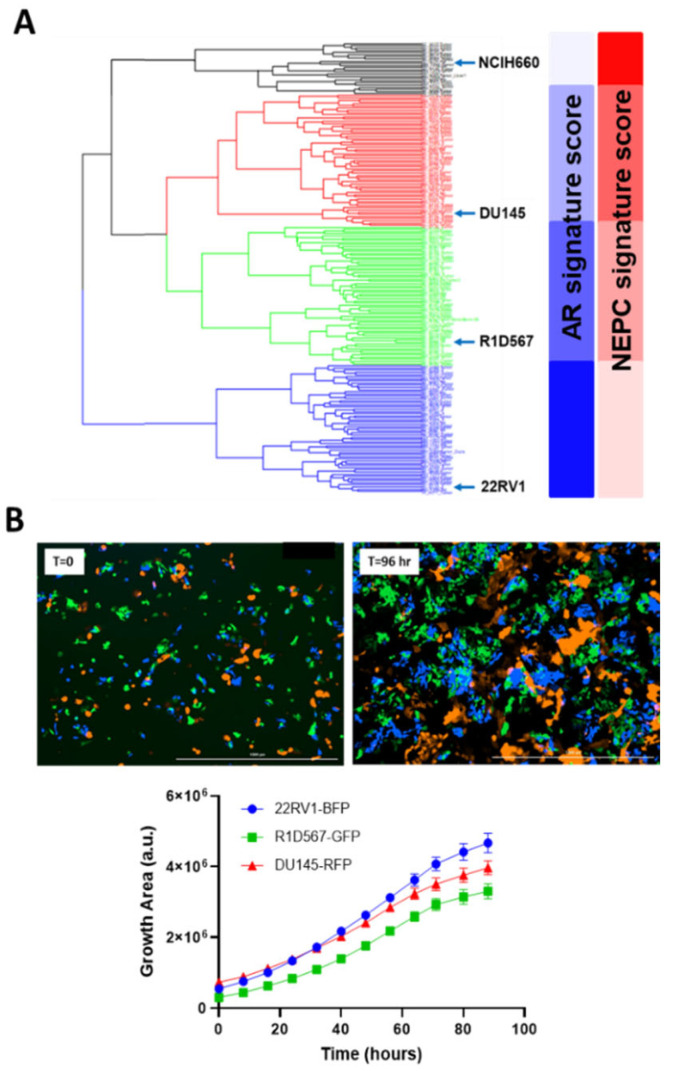
Development of mixed-cell model to represent intertumoral heterogeneity in CRPC tumors. (**A**) Hierarchical clustering of transcriptomes of 208 SU2C/PC-EC patient tumors and 10 prostate cancer cell-lines and their derivatives. Representative cell-lines selected from each cluster are highlighted. (**B**) Labeled PC cell-lines DU145-RFP, 22RV1-BFP, and R1D567-GFP proliferate exponentially in co-cultures in the absence of drug pressure over a growth period of 96 h.

**Figure 3 ijms-27-00413-f003:**
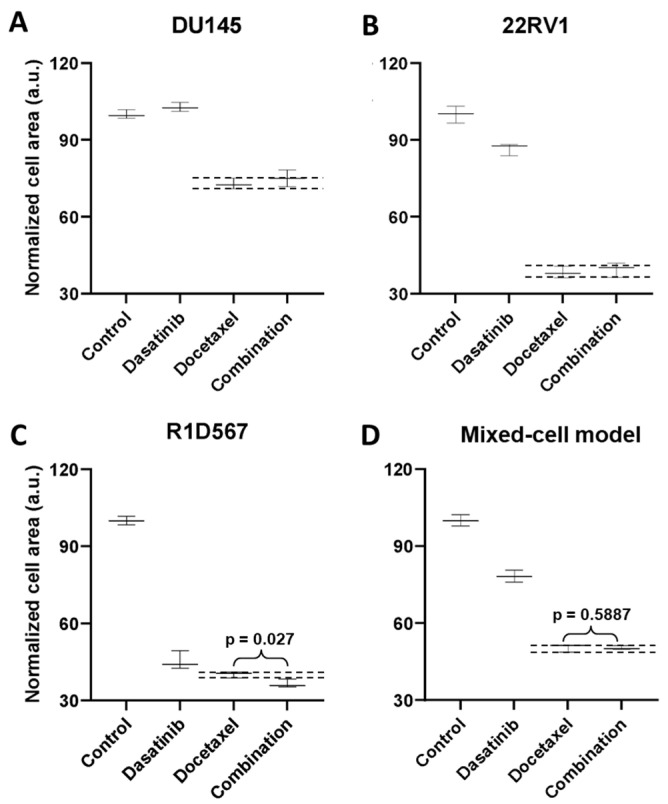
Mixed-cell models composed of DU145-RFP, 22RV1-BFP, and R1D567-GFP, depicting inter-patient heterogeneity, were treated with 0.4 μM dasatinib, 3 nM docetaxel, their combination, or the vehicle control (DMSO-containing media). Normalized growth areas of each component cell-line (**A**–**C**) were measured individually and totaled to determine the proliferation of the tumor mixture in the mixed-cell model (**D**) after 5 days of drug exposure. Black dashed lines outline the effects of the most efficacious monotherapy. Significance was assessed using the Mann–Whitney U test.

**Figure 4 ijms-27-00413-f004:**
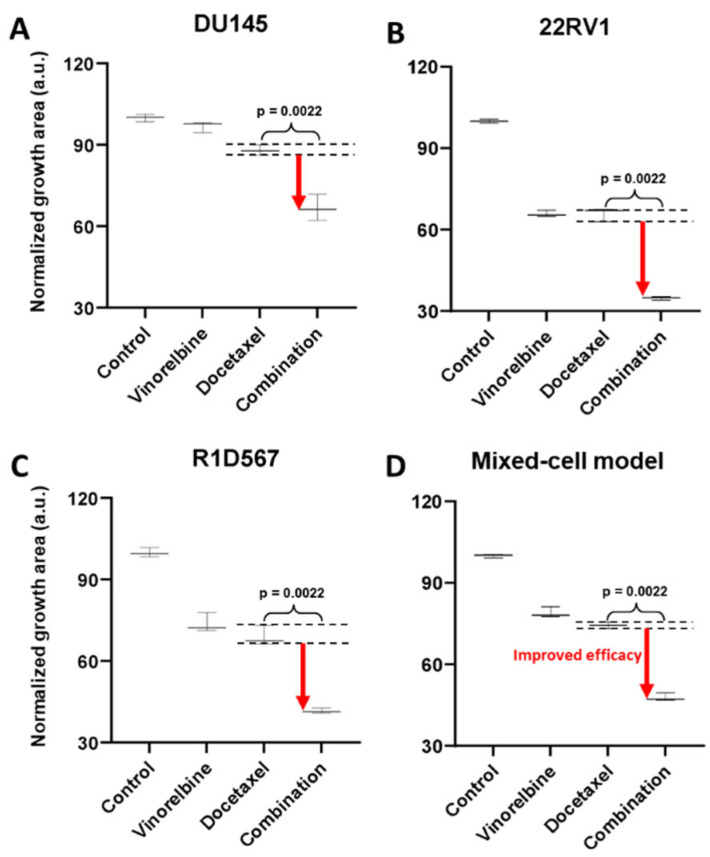
Mixed-cell models composed of DU145-RFP, 22RV1-BFP, and R1D567-GFP, depicting inter-patient heterogeneity, were treated with 8 nM vinorelbine, 3 nM docetaxel, their combination, or the vehicle control (DMSO-containing media). Normalized growth areas of each component cell-line (**A**–**C**) were measured individually and totaled to determine the proliferation of the tumor mixture in the mixed-cell model (**D**) after 5 days of drug exposure. Black dashed lines outline the effects of the most efficacious monotherapy. Significance was assessed using the Mann–Whitney U test.

**Figure 5 ijms-27-00413-f005:**
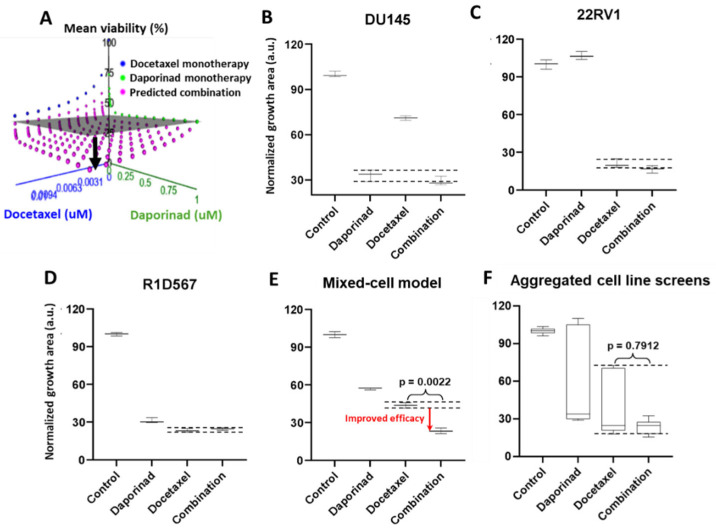
(**A**) IDACombo prediction of docetaxel and daporinad combination efficacy using monotherapy responses of 887 cancer cell-lines from CTRPv2. In (**A**), the gray plane denotes the highest average tumor growth inhibition achieved by either single drug in a collection of cancer cell-lines, and the pink dots represent the expected tumor growth when treated with the combination. When pink dots dip below the gray plane, the combination is expected to produce higher efficacy than the best single-agent treatment in a heterogeneous population. Mixed-cell models composed of DU145-RFP, 22RV1-BFP, and R1D567-GFP, depicting inter-patient heterogeneity, were treated with 5 nM daporinad, 3 nM docetaxel, their combination, or the vehicle control (DMSO-containing media). Normalized growth areas of each component cell-line (**B**–**D**) were measured individually and totaled to determine the proliferation of the tumor mixture in the mixed-cell model (**E**) after 5 days of drug exposure. Measured proliferation values across all component cell-lines were aggregated for each treatment condition to mimic common pharmacological practice (**F**). Black dashed lines outline the effects of the most efficacious monotherapy. Significance was assessed using the Mann–Whitney U test.

**Figure 6 ijms-27-00413-f006:**
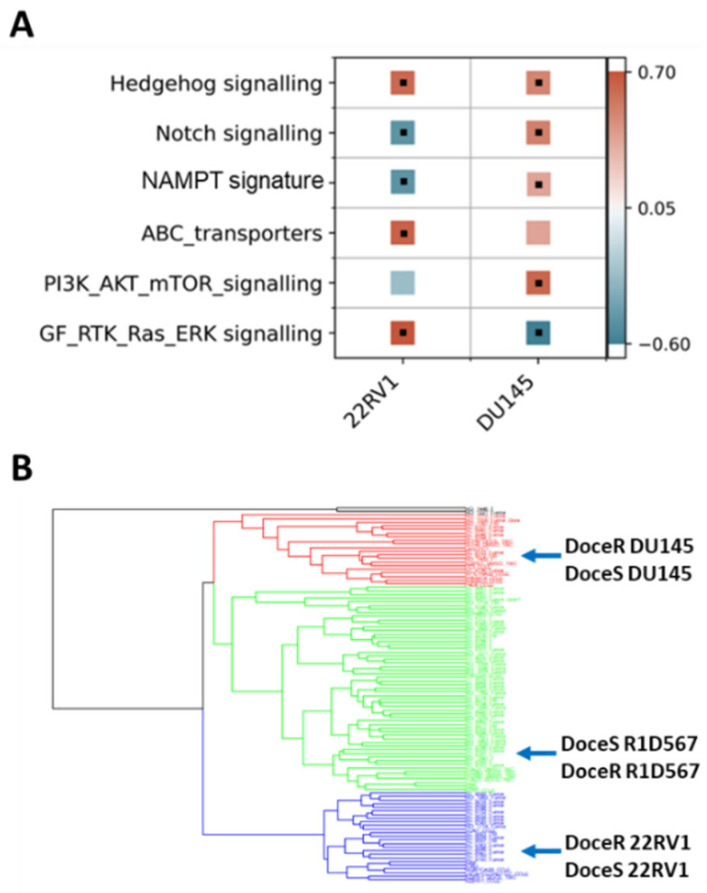
(**A**) GSEA gene set enrichment scores calculated using genes expressed differentially between docetaxel resistant (doceR) and sensitive (doceS) cell lines, separately in 22RV1 and DU145. Positive (red) and negative (blue) scores indicate upregulation and downregulation in the doceR derivative when compared to the parent doceS line, respectively. Black dots denote statistically significant differences with a corrected FDR q-value of less than 25%. (**B**) Hierarchical clustering of transcriptomes of 79 taxane exposed patient tumors from SU2C/PC-EC with taxane resistant and sensitive PC cell lines.

**Figure 7 ijms-27-00413-f007:**
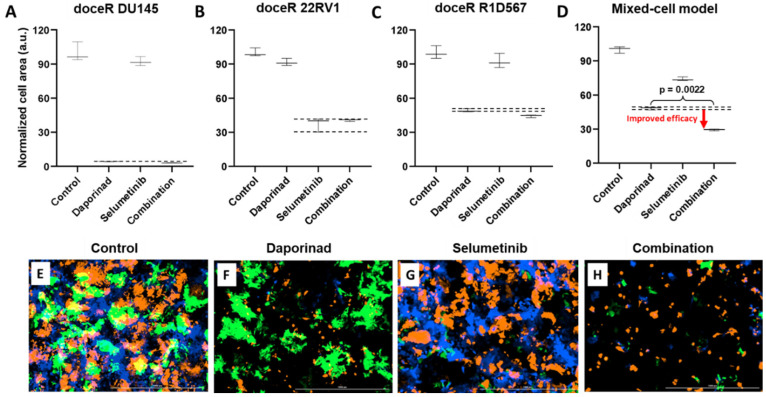
Mixed-cell model composed of docetaxel resistant DU145-BFP, 22RV1-GFP and R1D567-RFP, depicting docetaxel resistance in a heterogeneous patient population, were treated with 1 uM selumetinib, 4 nM daporinad, their combination, or vehicle control (no drug media). Normalized growth areas of the component cell lines (**A**–**C**) were measured individually and totaled to determine overall tumor proliferation after 5 days of drug exposure (**D**). Black dashed lines outline the effects of the most efficacious monotherapy. Significance was assessed using the Mann–Whitney U test. Representative microscopy images captured in 3 fluorescent channels, BFP, GFP and RFP are shown (**E**–**H**).

## Data Availability

The raw data generated in this study are available upon request from the corresponding author.

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
