# Peer review of "Novel Mixed Cancer-Cell Models Designed to Capture Inter-Patient Tumor Heterogeneity for Accurate Evaluation of Drug Combinations"

_ijms, 2025, doi:10.3390/ijms27010413_

Round 1
Reviewer 1 Report
Comments and Suggestions for Authors
Manuscript “Novel mixed cancer-cell models designed to capture inter-patient tumor heterogeneity for accurate evaluation of drug combinations” by Jena S et al, submitted to the International Journal of Molecular Sciences, discusses the importance of considering tumor heterogeneity in preclinical in vitro models to test new therapeutic modalities. The manuscript is beautifully written, and the authors have done a great job in explaining the methods and results very clearly.
I have the following suggestions and questions:
- I agree with the concept of the paper and the importance of involving tumor heterogeneity while designing assays for drug efficacy. However, the most challenging task is defining heterogeneity. For example, the neuroendocrine prostate cancer is one big class that was missed from the experiments due to difficulties because of non-adherent cell types (H660 or even LTL331). Also, non-tumor cells can play a role in efficacy. How authors plan to improve on these complexities. Mixed cell cultures would be better than single-cell models, but can this still address the very problem the authors recognized in the first place?
- 3 cell lines included in the model testing are DU145 (AR negative), 22Rv1 (AR and Arv7 expressing), R1D567 (Expresses Arv567es). However, many of the prostate tumors are only full-length AR-positive. Why was the only full-length AR-expressing line not included in the model? This is my biggest objection to the model system. Since authors are focusing on CRPC, AR-expressing CRPC models are also available, which should have been included in the model. Were those included in Hierarchical clustering analysis? The legend for Figure 2A suggests 10 PCa lines were included in the clustering. What were those cell lines?
- Figure 3A shows that DU145 exhibits a 70% growth area covered (approximately a 30% reduction on the arbitrary scale used) upon treatment with 3 nM docetaxel; however, Figure 4A shows that DU145 has around 90% growth area coverage with the same 3 nM Docetaxel dose. Similarly, in Figure 3B, 22Rv1 cells are reduced to around 30% with Docetaxel; however, the same cells show 60% on the scale in Figure 4 B. Why is there this big discrepancy in the growth inhibitory response between the 2 experiments? And the R1D567 shows growth inhibition with Docetaxel in Figure 3C, which is much higher than shown in Figure 4C. These are significant inter-experiment differences if we compare the response to the same drug, docetaxel, for the same cell line. If we believe growth inhibition shown by Docetaxel in Figure 3C is true, it will make findings in 4C not significant, as the bar for the docetaxel-treated R1D567 will fall to a similar level to the combination of Docetaxel and vinorelbine (Figure 4C and 4D)
- Since drug combinations are said to be effective at doses below IC50, IC50 values need to be included in the main manuscript.
- All three cell types differ in cell size, and I am sure fluorescent protein expression would also vary between cells (Expression of BFP vs. eGFP vs. TurboRFP). How authors plan to compensate for the differences in tag distribution among each cell type, because the intensity of tags (Fluorescent proteins) is being used to estimate the respective proportion in the mixed model. Is there a way to normalize this to the individual cell number, which I believe could serve as a gold standard for measuring the response to therapy?
- Additionally, there could be an issue due to spectral bleeding. Since images were acquired on the Cytation platform, does this have an option to compensate for the spectral bleeding of fluorescent proteins?
None
Author Response
Comments 1: I agree with the concept of the paper and the importance of involving tumor heterogeneity while designing assays for drug efficacy. However, the most challenging task is defining heterogeneity. For example, the neuroendocrine prostate cancer is one big class that was missed from the experiments due to difficulties because of non-adherent cell types (H660 or even LTL331). Also, non-tumor cells can play a role in efficacy. How authors plan to improve on these complexities. Mixed cell cultures would be better than single-cell models, but can this still address the very problem the authors recognized in the first place?
Response 1: The reviewer is correct that due to the requirement of specialized culturing conditions, we were unable to include the neuroendocrine subtype representing cancer cell line (NCIH660) in the current model of CRPC and as a result, lost the representation of patients from cluster 1 (colored in black) of the dendrogram in Fig. 2. However, we would like to note that cluster 1 is the least abundant patient cluster, constituting less than 10% of the cohort which implies that more than 90% of the cohort was duly represented in our model. Moreover, the main goal of this manuscript is to highlight the importance and feasibility of developing models that better represent disease heterogeneity and demonstrate proof of concept. While our models are not perfect, they are conceptually novel and enable validation of drug combinations in a heterogenous cell pool representative of a diverse clinical cohort, which has not been reported before. We believe that once published, our pipeline may be adapted and fine-tuned by other research groups to create better versions of the model. In the future, with the advancement of culturing techniques, more diverse and finicky cell-lines (both cancer and non-cancer cell types) can be incorporated into the mixed-cell models. Furthermore, we have added the following sentences in the Discussion section, page 10, paragraph 2: “. However, we were unable to include the NCIH660 cell line, representative of patient cluster 1, characterized by a high NEPC and low AR scores (colored black in the dendrogram in Fig. 2A), due to its requirement of specialized culturing conditions. Therefore, findings generated using the 3 cell line model on the efficacy of various drug combinations may not be applicable to CRPC patients with the neuroendocrine phenotype.”.
Regarding the role of the TME, we agree with the reviewer that the presence of non-cancer cell types will likely affect drug response in the tumor cells. In recent years, our investigations in multiple single-cell clinical datasets across different cancer types have shown that inter-patient diversity in the molecular composition of tumor cells is the main driver of variability in drug response. That is precisely the reason we chose to focus on the representation of tumor-centric inter-patient diversity in large clinical cohorts whose role in drug response is likely more pronounced than that of the TME. In the last paragraph of the discussion section of the manuscript, we have explicitly described our goals and the limitations associated with our model. In the future, additional work can and should be done to further improve them by incorporating additional cancer cell lines (representation of various cancer subtypes) as well as components of the tumor microenvironment to recapitulate interaction between cancer cells and TME.
Comments 2: 3 cell lines included in the model testing are DU145 (AR negative), 22Rv1 (AR and Arv7 expressing), R1D567 (Expresses Arv567es). However, many of the prostate tumors are only full-length AR-positive. Why was the only full-length AR-expressing line not included in the model? This is my biggest objection to the model system. Since authors are focusing on CRPC, AR-expressing CRPC models are also available, which should have been included in the model. Were those included in Hierarchical clustering analysis? The legend for Figure 2A suggests 10 PCa lines were included in the clustering. What were those cell lines?
Response 2: The process of selection of representative cell-lines for the models was entirely based on the transcriptomic similarity between prostate cancer cell lines and CRPC patient samples without an a priori consideration of AR status or other prostate specific biology. We hypothesized that a large transcriptomic signature is a more robust and comprehensive metric for patient tumor representation as opposed to specific markers such as AR status. The 10 prostate cancer cell lines included for this analysis were of diverse AR statuses, ranging from the early-stage androgen dependent (AR+) LnCap to the androgen independent (AR-) DU145 and NCIH660 lines. However, it should be noted that often progression of prostate cancer into the castration resistant stage is characterized by alterations acquired in the AR status of the PCa cells. These include loss of AR expression, emergence of constitutively active AR splice variants (AR-v7) and activation of AR-independent bypass pathways, all of which are exemplified by the selected representative cell-lines. The 10 prostate cancer cell lines included for clustering are: NCIH660, R1D567, R1AD1, BPH1, PC3, DU145, VCap, LnCap, 22RV1 and MDA PCa.
Comments 3: Figure 3A shows that DU145 exhibits a 70% growth area covered (approximately a 30% reduction on the arbitrary scale used) upon treatment with 3 nM docetaxel; however, Figure 4A shows that DU145 has around 90% growth area coverage with the same 3 nM Docetaxel dose. Similarly, in Figure 3B, 22Rv1 cells are reduced to around 30% with Docetaxel; however, the same cells show 60% on the scale in Figure 4 B. Why is there this big discrepancy in the growth inhibitory response between the 2 experiments? And the R1D567 shows growth inhibition with Docetaxel in Figure 3C, which is much higher than shown in Figure 4C. These are significant inter-experiment differences if we compare the response to the same drug, docetaxel, for the same cell line. If we believe growth inhibition shown by Docetaxel in Figure 3C is true, it will make findings in 4C not significant, as the bar for the docetaxel-treated R1D567 will fall to a similar level to the combination of Docetaxel and vinorelbine (Figure 4C and 4D).
Response 3: We appreciate the careful examination conducted by the reviewer and acknowledge the discrepancy in the measured responses to single-agent docetaxel therapy in the component cell-lines (Figs. 3 and 4). Significant inter-experimental differences in drug effects are sometimes observed due to the use of different batches/lots of the drug procured from the manufacturer, frozen stocks stored for different durations or precipitation, leading to variations in potency at the same concentration. In this case, while experiments pertaining to combinations of docetaxel with dasatinib and daporinad presented in Figs. 3 and 5 were conducted around February-March in 2023, experiments involving vinorelbine were done almost a year later in January 2024 (Fig. 4). Additionally, the docetaxel stocks used for these later experiments were obtained from a different manufacturer (Selleck Chemicals vs MedChemExpress) which might further explain the discrepancy observed. The Cell culture and reagents subsection in Materials and Methods has been accordingly updated. While the overall potency of docetaxel appeared to decrease in the later experiments with vinorelbine, we still observed similar trends in the relative efficacies of docetaxel in the component cell lines (as in similar efficacies in the 22Rv1 and R1D567 lines and much lower efficacy in DU145). Furthermore, the reduced potency did not alter our key findings and conclusions regarding the combination efficacy for which strong synergy between both the drugs was unambiguous. Also, in Figs. 3, 4 and 5, our goal is to compare responses across different treatment groups (single agent or combination therapies) within the same experiment. To ensure reproducibility and robustness of our results, a total of 6 technical replicates per treatment condition and 2 experimental replicates were generated and averaged.
Comments 4: Since drug combinations are said to be effective at doses below IC50, IC50 values need to be included in the main manuscript.
Response 4: Table S3 summarizing the IC50 values of all drugs investigated in this study (docetaxel, dasatinib, vinorelbine, daporinad and selumetinib) in the individual cell-lines, has been added to the supplementary file. These IC50 values were collected from a single experiment consisting of 6 technical replicates, conducted prior to the drug combination evaluations in the mixed-cell model to select for concentrations of the individual drugs to be used.
Comments 5: All three cell types differ in cell size, and I am sure fluorescent protein expression would also vary between cells (Expression of BFP vs. eGFP vs. TurboRFP). How authors plan to compensate for the differences in tag distribution among each cell type, because the intensity of tags (Fluorescent proteins) is being used to estimate the respective proportion in the mixed model. Is there a way to normalize this to the individual cell number, which I believe could serve as a gold standard for measuring the response to therapy?
Response 5: To avoid potential complications from differences in labeling efficacies between the component cell lines, we measured cell surface (growth) area in each fluorescent channel to determine proliferation of the respective cell-line instead of fluorescence intensity (see Methods: Longitudinal measurement of cellular proliferation in the mixed-cell model). Unlike fluorescence intensity which as the reviewer mentioned will vary based on the labeling efficiency, cell growth area is a more accurate measure of proliferation. The growth area of each component cell-line was measured at the start of treatment and after 5 days of exposure. For each well, the cellular area at the treatment endpoint was divided by the area recorded at the treatment start to normalize for differences in cell seeding densities across wells. Normalized growth areas were summed to determine growth/proliferation of the tumor mixture. Individual cell counts could not be reliably measured since the cells were labeled with a cytoplasmic tag instead of nuclear tag.
Comments 6: Additionally, there could be an issue due to spectral bleeding. Since images were acquired on the Cytation platform, does this have an option to compensate for the spectral bleeding of fluorescent proteins?
Response 6: We agree with the reviewer that spectral bleeding across channels will indeed affect our experimental measurements. That is precisely why we have carefully selected the labeling fluorophores to minimize spectral overlap. Moreover, imaging parameters such as the laser power/intensity, PMT gain and exposure time were kept low to minimize spectral bleed through when using the Cytation.
Reviewer 2 Report
Comments and Suggestions for Authors
The manuscript by Jena et al., “Novel mixed cancer-cell models designed to capture inter-pa- 2 tient tumor heterogeneity for accurate evaluation of drug com- 3 binations” The study is interesting but has some imitation. A functional analysis of key findings of this study is essential to approve its validity. Also, the co-cultivation of different cell lines together does not reflect tumor heterogeneity. Tumor heterogeneity is the diversity of cells within a tumor, which can exist between different patients in this is recognized as interpatient heterogeneity or within a single tumor that is recognized as intratumor heterogeneity. In this case, it is essential to determine the heterogeneity among the subpopulation of each cell line and the heterogeneity between the different cell lines. Also, what about the microenvironment tumor. As is widely reported and established, the crosstalk between the tumor and its microenvironment determines he resistance and response of the tumor to the applied therapeutics. Also, the immune microenvironment and its role in the context of tumor resistance and response. All these factors must be considered to develop a tumor heterogeneity model to functionally analyze tumor heterogeneity
Author Response
Comments 1: The manuscript by Jena et al., “Novel mixed cancer-cell models designed to capture inter-pa- 2 tient tumor heterogeneity for accurate evaluation of drug com- 3 binations” The study is interesting but has some imitation. A functional analysis of key findings of this study is essential to approve its validity. Also, the co-cultivation of different cell lines together does not reflect tumor heterogeneity. Tumor heterogeneity is the diversity of cells within a tumor, which can exist between different patients in this is recognized as interpatient heterogeneity or within a single tumor that is recognized as intratumor heterogeneity. In this case, it is essential to determine the heterogeneity among the subpopulation of each cell line and the heterogeneity between the different cell lines. Also, what about the microenvironment tumor. As is widely reported and established, the crosstalk between the tumor and its microenvironment determines he resistance and response of the tumor to the applied therapeutics. Also, the immune microenvironment and its role in the context of tumor resistance and response. All these factors must be considered to develop a tumor heterogeneity model to functionally analyze tumor heterogeneity.
Response 1: The goal and the novelty of this study is the representation of inter-patient tumor-centric heterogeneity, i.e. the diversity in tumor cells derived from different patients within a clinical cohort. During the process of drug development, cell lines for preclinical screening are often randomly selected based on availability and tested one at a time, prior to further in vivo and clinical testing. Since patient tumors are heterogeneous, the findings from these preclinical studies frequently fail to be translated clinically into diverse patient cohorts. In this study, we have developed and validated models that capture the inter-patient tumor heterogeneity in diverse clinical cohorts. Rationally selected cell-lines whose transcriptomes resemble distinct patient clusters (tumor variants) in a cohort were co-cultured to emulate the complex molecular landscape of the cohort. However, as stated by the reviewer, it is important to acknowledge that our current models do not capture the TME. Our rationale is that despite the new evidence of TME contributions to drug sensitivity, our results continue to support inter-patient differences in tumor molecular composition as the main driver of variability in drug response. Moreover, while there are substantial efforts in the field as of today to explore the role of TME, the impact of intrinsic inter-patient heterogeneity (genetic and phenotypic variability in tumor cells) has been largely overlooked, which remains the focus of our work. In the future, the inclusion of stromal components of the TME in our models will further improve on them and likely enhance their accuracy. We have included the discussion of these as limitations in the manuscript.
Round 2
Reviewer 2 Report
Comments and Suggestions for Authors
The manuscript is now improved and can be accepted for publication